# Paint Film Formation Characteristics on Conical Surfaces for Electrostatic Air Spray Painting

**Shuzhen Zhang \*** , **Jiongde Jin, Shijie Wu, Lun Jia and Xujie Ma**

School of Mechanical and Electrical Engineering, Lanzhou University of Technology, Lanzhou 730050, China; m13919113075@163.com (J.J.); 15769399917@163.com (S.W.); jialun2023@126.com (L.J.); 15101269219@163.com (X.M.)
\* Correspondence: zhangchen@lut.edu.cn

**Abstract:** When a curved target surface is coated in an electrostatic spray-painting process, the paint film pattern on curved targets will deform, and the thickness will change compared with the planar target due to the curvature characteristics of the target, the inhomogeneous electric field, and the flow field. Therefore, directly using the planar deposition distribution for painting the trajectory planning of curved surfaces causes large errors and low paint transfer efficiency. A study investigating the curved surface paint film thickness distribution characteristics is presented in this research to understand the relations among the target surface curvature, the electrostatic field, and the flow field distribution. Spray-painting process simulations were performed for conical surfaces of different curvatures. Moreover, the paint particle' trajectories, electric potential, and film deposition distribution were computed using ANSYS Fluent. The paint transfer efficiencies of these targets were computed. The results show that when the generatrix curvature of the conical surface increases gradually, the electric field intensity near the wall increases, which enhances the deposition of charged droplets on the wall. Moreover, the airflow field velocity increases as the curvature increases, which leads to a large diffusions of the spray flow field, so the paint thickness decreases, and the transfer efficiency is low.

**Keywords:** electrostatic air spray painting; conical surface target; film formation distribution; paint transfer efficiency; charged paint droplets

## 1. Introduction

Electrostatic spray painting is widely used for coating conductive substrates. It provides a nicely uniform film thickness with a reasonable paint transfer efficiency due to an additional electrostatic field supporting the droplets transported towards the target [1,2]. When the target geometry contains some complex surfaces such as large curvatures, sharp corners and recessed areas, the edge effect and the Faraday cage effect may occur in the electrostatic spray-painting process [3]. The angle and position of the paint droplets hitting a curved target are quite different from those of a planar target, and the spray pattern in planar workpieces will deform and the paint transfer efficiency will change; therefore, directly using the planar deposition model for trajectory planning will cause significant errors [4]. Different paint deposition methods have been proposed to obtain a more uniform paint thickness while planning the trajectory for curved surfaces.

Some scholars proposed projecting the plane paint film model onto a curved target for calculating the film deposition at any point, assuming that the droplets move along a straight line. Conner et al. [5] derived the projected planar deposition model on arbitrary surfaces using the concept of area magnification, as defined in differential geometry for the air atomizer spray painting the curved target. Xia et al. [6] adopted the projection theory to construct a deposition mode on a free-form surface using the curvature circle method for air spray painting. Their investigation also found that when the surface curves

away from the air atomizer, the geometric projection model is partially invalidated and the surface curvature has a significant impact on the actual deposition pattern on the curved target. Ye [7] presented a numerical simulation of the spray-painting process using a pneumatic sprayer and investigated the sensitivity considering different operating conditions and complicated target geometries. Chen [8] adopted the Euler–Euler approach in computational fluid dynamics (CFD) theory to simulate the paint thickness distribution for painting curved surfaces of an outer cylindrical and an inner cylindrical surface using a pneumatic atomizer. Chen et al. [9] took spherical surfaces with different diameters as the target surfaces, established the gas–liquid two–phase flow coupling process of air spray painting using the Euler–Lagrange method, and provided the spraying film thickness distribution model of the spherical surface. Osman et al. [3] simulated and analyzed the electrostatic rotary bell spray-painting process on a target with perpendicular protrusions or depressions to explore the influence of particle size, protrusions height or depressions depth, corner radius, and other factors on the coating thickness distribution, but did not use experiments to verify the calculations. Toljic [10] created a full 3D numerical model of the electrostatic coating process on a moving target using the CFD program FLUENT. The simulated target took the shape of a car door, assuming a specified hole on the surface of the plate for the handle. Their numerical results showed that the deposition around the door handle was at a higher level due to the electrostatic edge effect. Liu [11] established a double bias β film thickness distribution model for electrostatic rotary cup spray painting and constructed a curve projection model based on a normal distribution combined with experimental data. Guettler [12] proposed an injection model and a framework to calibrate the injection model coefficient via a metamodel-based optimization in combination with experimental painting-specific data. Afterwards, a simulation with a complex workpiece including moving atomizers along a robot path was simulated with the optimized injection parameters. Although numerical modeling of the electrostatic spraying process has been used by a number of researchers to solve the electrostatic spray-painting process and film thickness deposition model for robot spray painting trajectory planning [12–17], understanding the effect of surface characteristics, including shape and curvature, on paint deposition and paint transfer efficiency has not been modeled and investigated extensively. Therefore, it is necessary to combine the geometric characteristics of the target surface with the sprayer operating parameters to study the film thickness distribution in the electrostatic air spray-painting process, in order to provide an accurate prediction for the film thickness distribution with high paint transfer efficiency.

Gaussian curvature, an intrinsic metric of the surface, is solely determined by the geometric properties of the surface. It is defined as the product of the principal curvatures $k_1$ and $k_2$ at a point of the surface and quantifies the bending degree of the surface [18]. A low Gaussian curvature indicates relative flatness, while a high Gaussian curvature suggests significant curvatures or irregularities. Considering that the principal curvatures at a point on the conical surface $k_1 = 0$ and $k_2$ changes continuously along its generatrix, this study will take conical surfaces with different cone angles as the sprayed target to analyze the electrostatic air spray-painting process in order to explore the coupling effects of electrostatic field and flow field, as well as curvature changes on the paint deposition distribution for curved surfaces.

## 2. Paint Droplets Transport Model

### 2.1. Approach

An electrostatic field is generated by applying a voltage differential between the sprayer and the earthed workpiece target. The discrete paint droplets are produced by the pneumatic atomizer and charged due to free ions produced from corona discharge at the electrode. With the interaction of the electrostatic force, a strong airflow coaxial with the sprayer, as well as gravity, the charged droplets are pushed toward the target [2]. In particular, aerodynamic and electrical forces play an equally important role in the transport process of the droplets [12–15]. The spray-painting process is characterized as a two-phase

flow of air and fluid, and coupling multi-field. The flow field and electrostatic field are computed using an Eulerian approach, and the droplet's motion is computed using a Lagrangian that investigates the motions of each fluid particle, tracking their entire trajectory and capturing the evolution of their physical properties throughout the process [12–15]. The air is a continuous phase that is modeled using the Eulerian conservation equation for mass and momentum [13,15]. The turbulence effects are included in the realizable $k$–$\varepsilon$ model [13,15], which implies that the model satisfies specific constraints on the Reynolds' stresses that make the model more consistent with the physics of turbulent flow.

*2.2. Electrostatic Field*

The electrostatic force deeply affects the motion of the charged paint droplets, thus, we are required to solve the electrostatic field model with very high precision. The electric field with space charge can be described by Maxwell equation and the conservation equation [14] for the flow of the ions, as follows:

$$\nabla^2 \varnothing = -\frac{\rho}{\varepsilon_0} \tag{1}$$

$$E = -\nabla \varnothing \tag{2}$$

where $\varnothing$ is the potential function, $\rho$ is the space charge density, $\varepsilon_0$ is the permittivity of a vacuum and $E$ is the electric field intensity. The above equations are realized using a Fluent user-defined scalar (UDS) equation, which is a generic transport equation for a passive scalar, and a Fluent user-defined function (UDF), which is an additional subroutine developed to compute the electrostatic field. A Lagrangian particle tracking model is coupled with the electric field in the iterative process.

*2.3. Motion Trajectory of the Charged Droplets*

Considering the coupling of the airflow with the droplets and the effect of the space charge on the droplet's charge, the electrostatic field is computed using the Fluent MHD model. The motion trajectory of the charged droplets with identical polarity is computed using a discrete phase model (DPM), which is a Lagrangian model for tracking the particles or droplets using the computational domain where the solution of the continuous phase has been solved using an Eulerian approach. A DPM model tracking droplets mainly includes a series of differential equations computed using a Lagrangian approach [14,16]. The trajectory of the droplets is calculated using the integration of motion equations [12–15], as the following:

$$m_p \frac{du_p}{dt} = m_p \frac{u - u_p}{\tau_r} + m_p \frac{g(\rho_p - \rho)}{\rho_p} + F_E \tag{3}$$

$$F_E = \frac{q}{m} E \tag{4}$$

where $u$, $u_p$ are the airflow and the droplet velocity vector, respectively; $\tau_r$ is the drag coefficient; $g$ is the acceleration of gravity; $\rho_p$ is the density of the paint droplet; $m$ and $q$ are the mass and the charge of individual paint droplets, respectively; and, $F_E$ is the electric field force.

**3. Numerical Simulation Setup**

*3.1. The Computational Domain*

The computational domain of electrostatic air spray painting is a rectangular box of 300 mm on a side that completely encloses the sprayer and the target. The target is a partial conical surface and is placed at a distance of 250 mm from the sprayer electrode needle. The investigations shown here were carried out using a GRACO Pro XpTM electrostatic air sprayer. A picture of the sprayer, including the electrode, the shaping air holes, the pint hole, the atomizing air hole, as well as the assisting air holes is depicted in Figure 1a. The geometric model of the electrostatic air sprayer cap is constructed based on the measurements shown in Figure 1b.

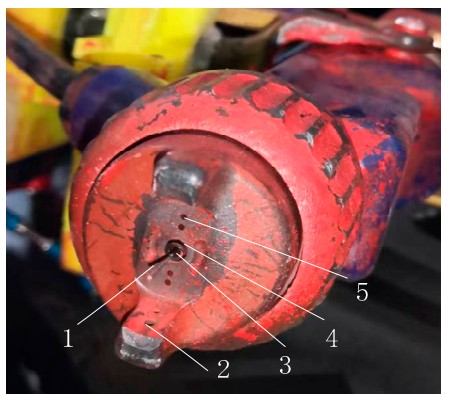

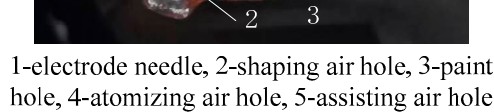

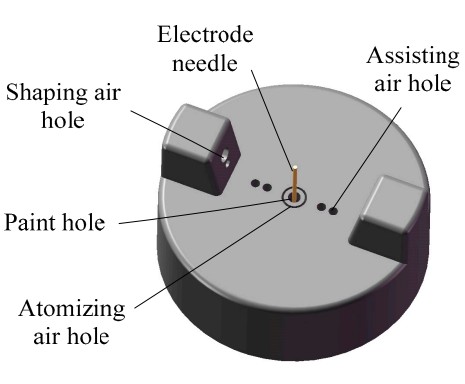

1-electrode needle, 2-shaping air hole, 3-paint hole, 4-atomizing air hole, 5-assisting air hole

(**a**)　　　　　　　　　　　　　　　(**b**)

**Figure 1.** Electrostatic air sprayer: (**a**) electrostatic air sprayer for research; (**b**) geometric model of the electrostatic air sprayer cap.

The flow field and the electrostatic field are solved using finite volume methods of ANSYS19.0/Fluent software. A hybrid unstructured mesh with 1,400,000 cells is used in the computational domain, a high-density mesh is close to the electrode needle and all holes in the region where the higher flow gradients and higher droplet concentrations are resolved properly. The computational domain and electrostatic air sprayer cap mesh distribution are shown in Figure 2.

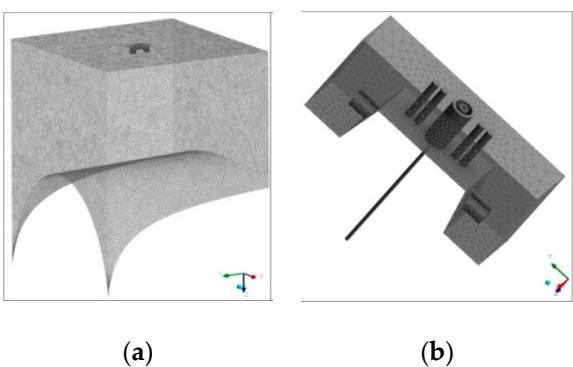

(**a**)　　　　　　　　　　　(**b**)

**Figure 2.** Mesh distribution of computational domain of the conical target: (**a**) computational domain of the conical target; (**b**) mesh distribution of the electrostatic sprayer air cap.

The targets in the study are the conical surfaces with a length of 500 mm generatrix having $80°$, $100°$, and $120°$ cone angles, respectively. The center of the paint film distribution is at the center point of a generatrix with principal curvatures of $k_1 = 4.78 \times 10^{-3}$, $k_1 = 3.29 \times 10^{-3}$, and $k_1 = 2.28 \times 10^{-3}$, respectively, and all $k_2$ are 0.

### 3.2. Continuous Phase

For the continuous air phase, the four side boundaries of the domain are set as constant pressure surfaces at 0 Pa and 293 K. The top boundary is set using a mass flow inlet boundary. The pressures of the shaping air of 0.04 MPa, the assisting air of 0.03 MPa, and the center atomizing air of 0.2 MPa are used. The boundary conditions for the electrostatic potential field are specified in terms of a fixed potential value of $-60$ kV at the electrode needle, while the conical target is grounded (0 V). All of the other solid surfaces, such as the sprayer, are insulated, and, therefore, a normal gradient of 0 V is specified. The major application parameters are summarized, as shown in Table 1.

**Table 1.** The parameters of continuous phase.

| Paint Viscosity | Paint Density | Paint Surface Tension | Paint Electrical Conductivity | Air Density | Air Viscosity |
|---|---|---|---|---|---|
| $0.08 \, \text{kg}/(\text{m}\cdot\text{s})$ | $1200 \, \text{kg}/\text{m}^3$ | $71.9 \, \text{mN}/\text{m}$ | $5 \times 10^{-4} \, \text{mS}/\text{m}$ | $1.225 \, \text{kg}/\text{m}^3$ | $1.789 \times 10^{-5} \, \text{kg}/(\text{m}\cdot\text{s})$ |

*3.3. Discrete Phase*

The spray-painting process simulations in previous studies were generally based on the assumption that liquid instantly breaks up after leaving the nozzle on account of the complexity of the liquid atomization process [19,20]. The experiments obtained the size distributions of the droplets, which are time-consuming and expensive. Uniform droplets with an initial diameter of 65 μm [8] were injected into the airflow at the position of the paint hole in this research. The Taylor analogy breakup (TAB) model was used to predict the secondary breakup and child droplet diameters. The number of computational droplets plays an important role in the Lagrange tracking method; 90,000 droplets per DPM calculation were used, which can be proven to be accurate enough for the current application. The paint hole was set as the mass inlet, mass flow rate was $1.32 \times 10^{-3} \, \text{kg}/\text{s}$, and the initial velocity was $10 \, \text{m}/\text{s}$.

**4. Results and Discussion**

The results are divided into two sections. The first section is for the results of the continuous phase, i.e., the flow field and the electrostatic field. The second section shows the results for the droplet trajectories, the transfer efficiency, and the film thickness distribution.

*4.1. Continuous Phase*

The main forces that determine the transport of the paint droplets are the electrostatic force and the drag force, with gravity being a minor contribution in the domain. The air velocity flow fields through the XOZ plane for different targets are shown in Figure 3.

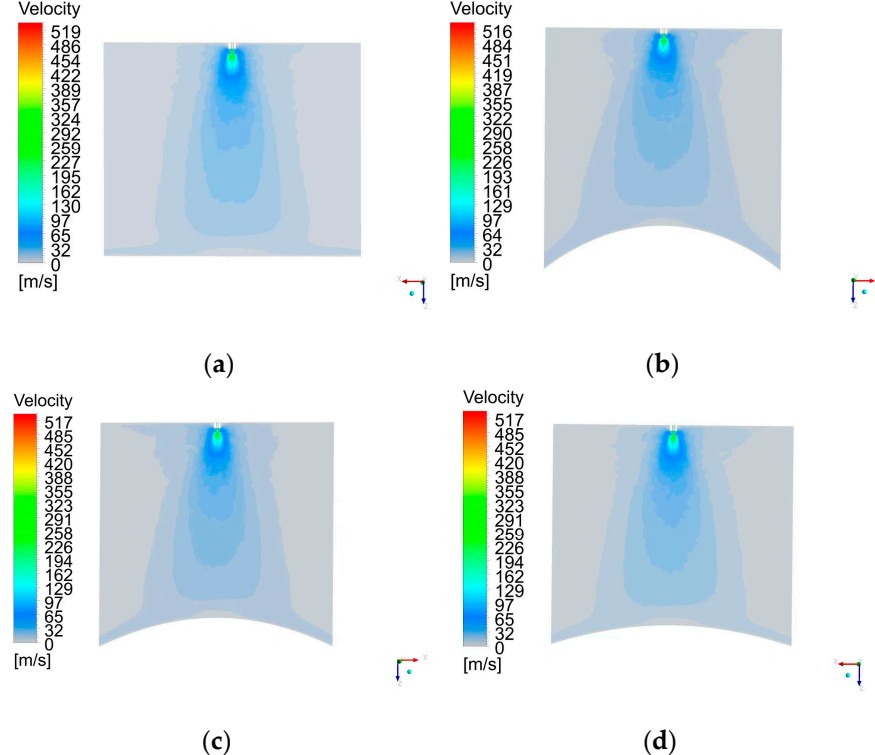

**Figure 3.** Air velocity contours for different conical targets: (**a**) $k_1 = 0$; (**b**) $k_1 = 4.78 \times 10^{-3}$; (**c**) $k_1 = 3.29 \times 10^{-3}$; (**d**) $k_1 = 2.28 \times 10^{-3}$.

In Figure 3a, the airflow sharply turns sideways near the target surface in the film-forming region, and the z-direction velocity rapidly slows down. Figure 3b–d shows that the airflow spreads along the curve direction of the conical surface and gradually increases as the curvature increases, which contributes to some paint droplets easily overcoming the viscosity and splashing out. The air velocity distribution for spray painting a conical target is similar to that for spray painting a spherical surface by Chen et al. [9] and for a cylindrical surface by Xie et al. [21] using an air spraying gun. However, in Figure 3, a radially outward convected region surrounding the air cap can be observed due to the influence of the electrical field on the airflow.

The electrostatic field's electrical potential distribution in the XOZ section for different targets is shown in Figure 4.

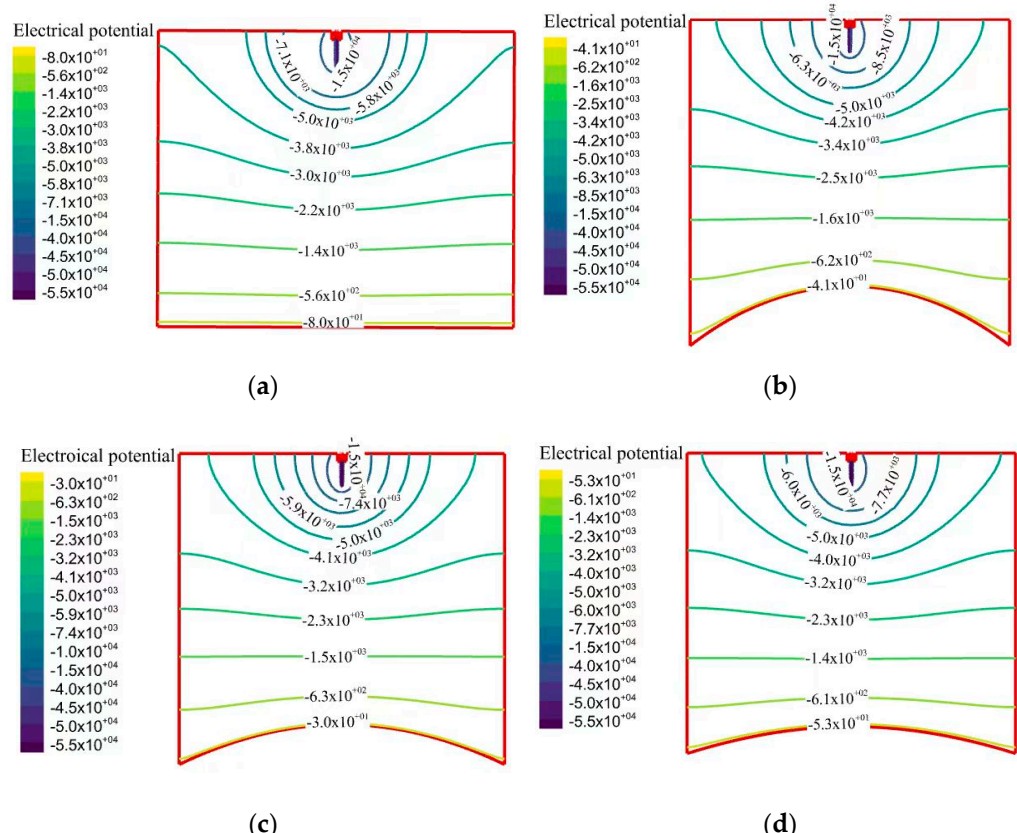

**Figure 4.** Potential field contours on the center plane in the region between the sprayer and different curvature targets: (**a**) $k_1 = 0$; (**b**) $k_1 = 4.78 \times 10^{-3}$; (**c**) $k_1 = 3.29 \times 10^{-3}$; (**d**) $k_1 = 2.28 \times 10^{-3}$.

The intensity of the electric field is inversely proportional to the separation distance between the potential lines. As shown in Figure 4, regions with high electric strength exist in the vicinity of the electrode needle, where the aerodynamic force also plays an important role. The electrical potential near the targets with different curvatures is different. In Figure 4a, the equipotential lines close to the sprayer and to the electrode tend to follow the outline of the geometry, and, away from the sprayer surface, the lines become more spherical in shape, resembling the solution of the electrostatic field for targets with different curvatures, as shown in Figure 4b–d. This result is consistent with that obtained by Viti [14] in an electrostatic rotary bell spray-painting process; the equipotential lines close to the sprayer in this research are long and elliptical, which for Viti [14] are cup-shaped, due to different discharge of the spray gun. When approaching the target surface, the potential field tends to follow the target outline to meet the condition of 0 V on the target. The potential curve coaxially near the sprayer is denser than that away from the coaxial, and the density increases with increasing cone curvature, indicating that the electrostatic field intensities at these locations are higher than those coaxially away from the sprayer.

### 4.2. Discrete Phase

In the spray-painting process, the droplets are transported to the target by the jet stream and the electric field. The droplet trajectories are calculated by solving Equation (3). The generated particles with trajectories smaller than or equal to 80 μm are visualized. The droplet size distributions are shown in Figure 5.

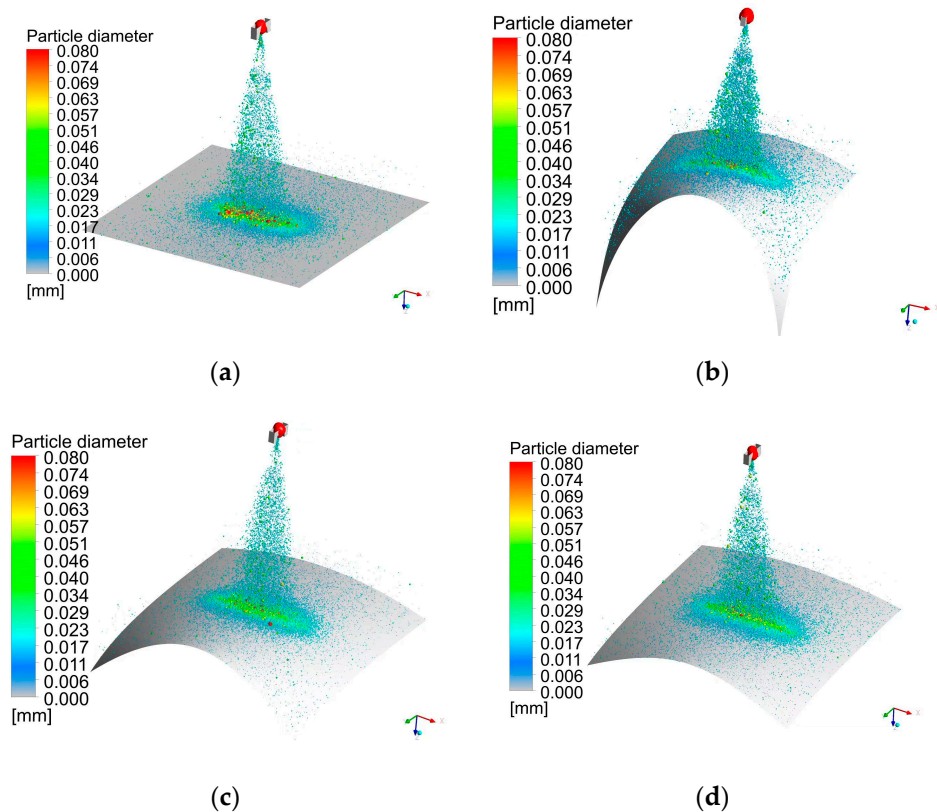

**Figure 5.** Paint droplet size distributions for sprayed targets of different curvatures: (**a**) $k_1 = 0$; (**b**) $k_1 = 4.78 \times 10^{-3}$; (**c**) $k_1 = 3.29 \times 10^{-3}$; (**d**) $k_1 = 2.28 \times 10^{-3}$.

Figure 5 shows that larger paint droplets, due to their high inertia, are less affected by the shaping airflow. As a result, they tend to concentrate towards the center of the spray cone. On the other hand, the smaller droplets are easily influenced by the airflow and deviate from the spray cone, spreading outwards. Therefore, the bigger and medium-sized droplets adhere effectively to the target surface. This result is in agreement with that obtained by Chen et al. [9] in an air spray-painting process. The droplets outside the spray cone tend to escape the shaping air and form an elliptical disk around the sprayer; the lengths of the long shaft and short shaft of the ellipse are determined by the momentum of the droplets. Figure 5a shows that when the target is in the plane, larger particles adhere to the center of the paint film. In Figure 5b, the sprayed surface has the highest curvature, while there are the fewest large and medium-sized droplets deposited on the surface. Figure 5c shows that more large and medium-sized droplets compared to Figure 5b are deposited on the sprayed surface with the second highest curvature, In Figure 5d, the sprayed surface has the lowest curvature, but there are still more large and medium-sized droplets compared to Figure 5c, second only to deposition on the plane shown in Figure 5a. Figure 5 shows that more particles spread along the curved direction near the conical surface, and the number of larger particles adhering to the center gradually decreases as the curvature of the conical target increases.

### 4.3. Paint Film Deposition Thickness Distribution and Transfer Efficiency

The film deposition thickness contour for different conical targets is shown in Figure 6. A narrow elliptical film deposition is formed with an extension along the X-direction. This contour result is similar to what was obtained by Chen et al. [9], Xie et al. [21], and Ye [15] in an air spray-painting process, whereas the film thickness for an electrostatic air spray-painting process is higher than that for air spray painting due to the electric field between the sprayer and the target. As the target curvature increases, so does the extension of the elliptical film's long axis, but the length of the short axis decreases. The maximum vertical coordinate representing the film thickness in Figure 6a is 37.77 μm. In Figure 6b, it reaches 32.65 μm. In Figure 6c, it is 34.62 μm. In Figure 6d, it is 36.47 μm. The maximum vertical coordinate representing film thickness gradually decreases as the curvature of the targets increases. The paint thickness profiles for the plane and conical targets are self–similar in shape, whereas the maximum thickness occurs for the plane target. These results also highlight the high paint transfer efficiency for the flat target, which is discussed next. The film distribution parameters are shown in Table 2. The maximum film thickness at the center of the deposition on the X and Y sections decreases gradually as the curvature increases. Figure 7 shows the paint transfer efficiencies computed using the Fluent DPM model.

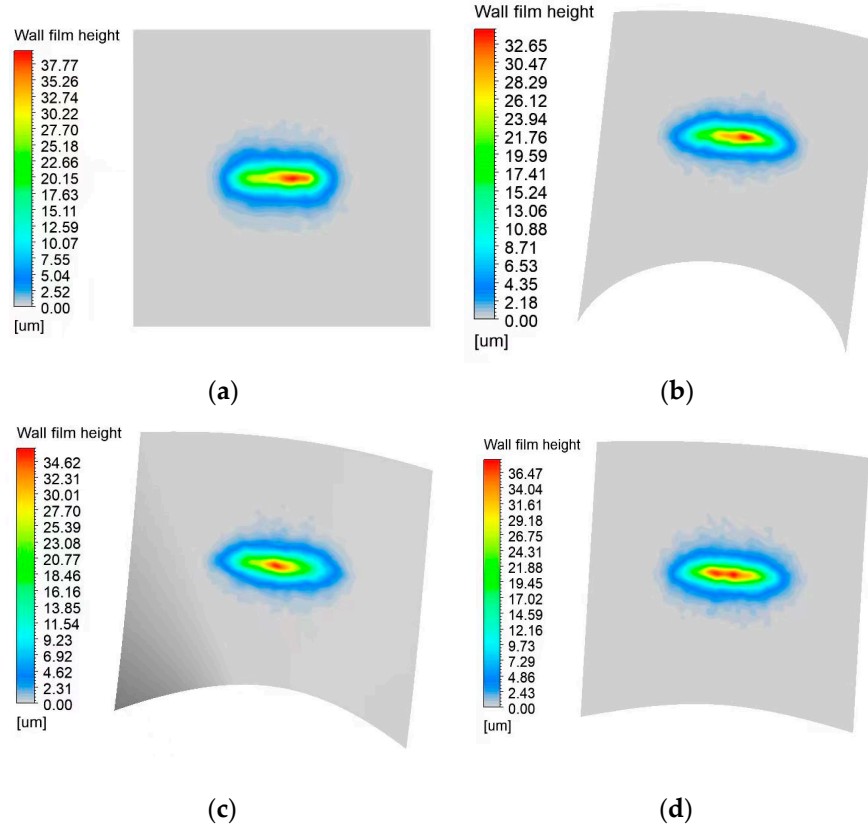

**Figure 6.** Film deposition thickness contours on different conical targets (**a**) $k_1 = 0$; (**b**) $k_1 = 4.78 \times 10^{-3}$; (**c**) $k_1 = 3.29 \times 10^{-3}$; (**d**) $k_1 = 2.28 \times 10^{-3}$.

The paint transfer efficiency is relatively high for the planar target and decreases as the conical target's curvature increases. Due to the curved characteristics of the cone surface along the radial direction, the velocity along this direction increases in the flow field near the wall. Some relatively small droplets can easily overcome the electric field force, paint viscidity, and surface tension to splash. As a result, the droplets diffuse so largely that the effective film-forming range is small, and the paint transfer efficiency decreases along with increasing cone curvature. When the principal curvatures of the center point of the

target's paint film are $k_1 = 0$, $2.28 \times 10^{-3}$, $3.29 \times 10^{-3}$, and $4.78 \times 10^{-3}$, the paint transfer efficiencies are 84.69%, 69.04%, 64.19%, and 48.53%, respectively.

**Table 2.** Film thickness distribution parameters on X and Y sections of conical targets.

| Curvature of Conical Targets | Maximum Thickness/µm | Maximum Thickness on X Section/µm | Maximum Thickness on Y Section/µm |
|---|---|---|---|
| 0 | 39.588 | 39.2 | 35.2 |
| $2.28 \times 10^{-3}$ | 38.223 | 38.4 | 33.7 |
| $3.29 \times 10^{-3}$ | 36.285 | 35.5 | 31.8 |
| $4.78 \times 10^{-3}$ | 34.215 | 31.9 | 30.3 |

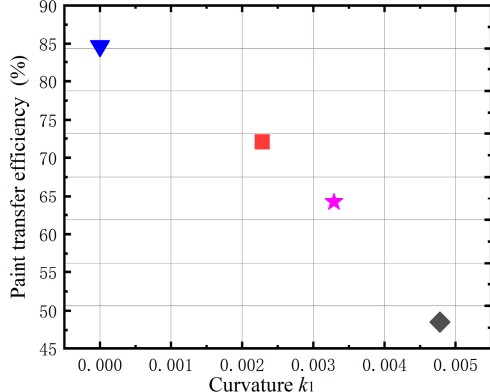

**Figure 7.** The paint transfer efficiency for conical targets with different curvatures.

The velocity increase in the flow field has a reducing effect on the film thickness. In the near-wall region of the sprayed target with different curvatures, the electrical potential distribution is similar, whereas the airflow tends to diffuse with an increase in curvature. The paint droplets diffuse with the airflow, resulting in a decrease in the paint deposition thickness, indicating that the flow field plays a major role near the curved target.

## 5. Experimental Investigation

To verify the simulation results, electrostatic air spray painting experiments were conducted. The electrode voltage was set as −60 kV. The center of the target was placed at a distance of 250 mm away from the sprayer nozzle. Each spray-painting process lasted for 1 s, as measured using a stopwatch.

Figure 8 shows the experimental paint films deposited on the different conical surfaces. Figure 8a shows the elliptical profile paint film sprayed on the plane, while Figure 8b–d show narrow elliptical paint film profiles sprayed on the conical surfaces. There is good agreement between the experiment and simulation results for the film distribution patterns.

After the paint film was solidified, the thickness was measured using a film thickness gauge along the long and short axes of the film distribution at an interval of 10 mm. The YUWESE EC-770 coating thickness gauge, shown in Figure 9, was utilized in the experiment to quantify the thickness of the coating. The precision of the gauge was 1 µm within a range of 0–999 µm. The film thickness at each point was measured five times, and the average was used as the final thickness value. Figure 10 shows the comparison between the measured and simulated film thicknesses.

The simulated film thickness distributions are drawn in body-fitted coordinates along the X- and Y-axes of elliptical film deposition. The film thickness along the X-axis is almost symmetrical about the center point, except on the plane target and offset along the Y-axis, where these offsets appear to result from the lateral airflow in the experiment. Figure 10a,b show that the film thicknesses of different target surfaces along the X-axis are almost higher than those along the Y-axis. These simulated film thickness distributions are always

slightly higher than the experimental values. The difference may be due to measurement errors. The film thickness of the plane is higher than that of the curved surface, and the thickness decreases as the target surface's curvature increases whether it is in experiment or simulation. This result is in agreement with that obtained by Chen et al. [9] in an air spray-painting spherical surface process, whereas the film thickness for electrostatic air spray painting is higher than that for air spray painting due to the electric field between the sprayer and the target.

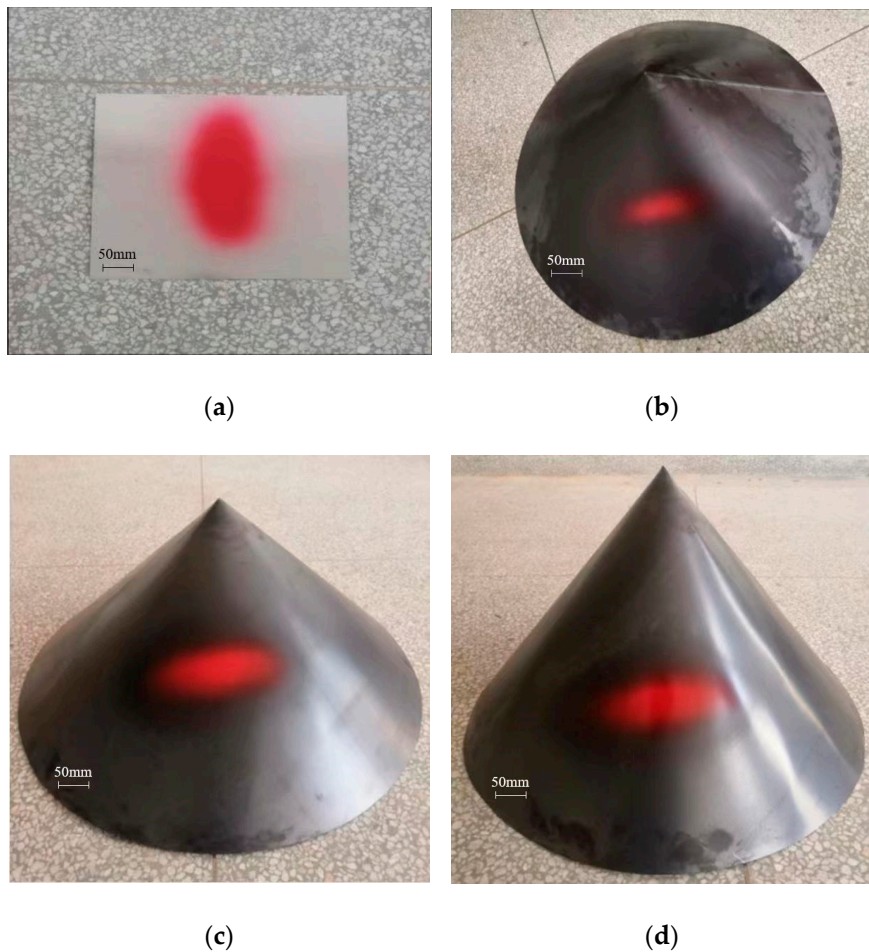

(a)                                    (b)

(c)                                    (d)

**Figure 8.** Experimental paint distributions on conical targets with different curvatures: (**a**) $k_1 = 0$; (**b**) $k_1 = 2.28 \times 10^{-3}$; (**c**) $k_1 = 3.29 \times 10^{-3}$; (**d**) $k_1 = 4.78 \times 10^{-3}$.

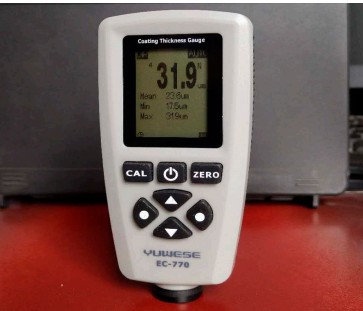

**Figure 9.** Coating thickness gauge for this experiment.

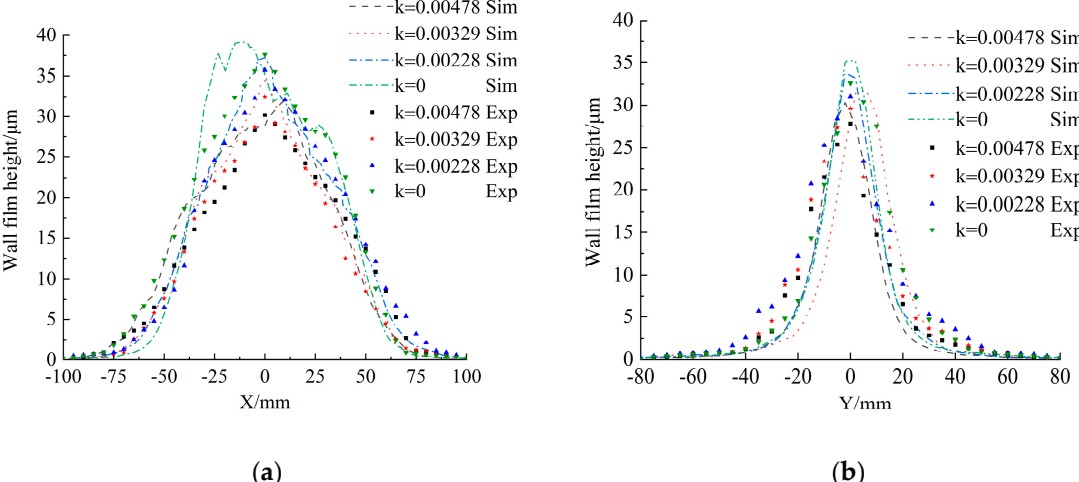

**Figure 10.** Film thickness distributions for different conical surfaces: (**a**) film thickness along the X-axis; (**b**) film thickness along the Y-axis.

## 6. Conclusions

A numerical simulation of the electrostatic spray-painting process using an air atomizer with external charging was presented. The numerical model accounts for the effects of aerodynamic and electrostatic forces, as well as the turbulence dispersion on the paint droplets, the two-phase coupling effect, and the effects of the target's geometrical characteristics on the airflow field and electrostatic field. It was found that the effects of the target's geometrical characteristics on the film thickness distribution could not be neglected.

(1) Compared with the conical surface sprayed, the maximum film thickness on the planar target is the greatest. As the curvature of the conical target increases, the maximum film thickness decreases, and the paint film distribution range becomes narrow.

(2) For a curved sprayed target, the airflow diffuses the fastest along the direction of maximum curvature on the target surface. With an increase in the curvature of the conical target, the flow field velocity near the wall region of the sprayed surface increases, and the number of small droplets carried by the airflow increases, while the number of droplets deposited on the target decreases, resulting in a lower paint transfer efficiency.

(3) With an increase in the curvature, except for the spray center, the electric potential lines in the near-wall region are sparser, and the electric field force acting on the charged paint droplets is weakened. The paint transfer efficiency decreases with an increase in the curvature of convex conical surfaces.

**Author Contributions:** S.Z. conceived the concept and wrote the paper; J.J. carried out the simulation; X.M. and L.J. analyzed the data; S.W. and X.M. performed the experiment. All authors have read and agreed to the published version of the manuscript.

**Funding:** This research was funded by the National Natural Science Foundation of China, grant number 52265065.

**Institutional Review Board Statement:** Not applicable.

**Informed Consent Statement:** Not applicable.

**Data Availability Statement:** The data that support the findings of this study are available from the corresponding author, upon reasonable request.

**Conflicts of Interest:** The authors declare no conflict of interest.

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
