# Peer review of "Paint Film Formation Characteristics on Conical Surfaces for Electrostatic Air Spray Painting"

_coatings, doi:10.3390/coatings13101808_

Round 1

Reviewer 1 Report

The aim of the article is missing, and novelty is not exposed. It must be added in section1.

If the model is self developed, it should be presented and explained more in detail. In present form it is not clear enough. Improve all subsections 2.1, 2.2, 2.3. Add some more references

L100: “…computed using a Lagrange…” please explain how,  more in detail?

L137-L141: insert space between number and unit, e.g. 300 mm

L161-L162: insert space between number and unit

Table 1: insert space between number and unit

L225: insert space between number and unit 80 µm

Check Figure 7 value K: is  capital letter correct?

Author Response

Response to Reviewer 1 Comments

1. Summary

Thank you very much for taking the time to review this manuscript. Please find the detailed responses below and the corrections highlighted.

2. Questions for General Evaluation

Reviewer’s Evaluation

Response and Revisions

Does the introduction provide sufficient background and include all relevant references?

Yes/Can be improved/Must be improved/Not applicable

We have added the aim of article in section 1.

Are all the cited references relevant to the research?

Yes/Can be improved/Must be improved/Not applicable

We have add some more cited references in corresponding subsections.

Is the research design appropriate?

Yes/Can be improved/Must be improved/Not applicable

The design of the research has been repeatedly analyzed and verified by the experiments.

Are the methods adequately described?

Yes/Can be improved/Must be improved/Not applicable

Are the results clearly presented?

Yes/Can be improved/Must be improved/Not applicable

Are the conclusions supported by the results?

Yes/Can be improved/Must be improved/Not applicable

3. Point-by-point response to Comments and Suggestions for Authors

Comments 1: The aim of the article is missing, and novelty is not exposed. It must be added in section1.

Response 1: Thank you for pointing this out. We agree with this comment. Therefore, we have added the aim of article in section 1, this change can be found – page 2, line 82-93. And the aim of article is also mentioned in page 1, line 72-81.

Comments 2: If the model is self developed, it should be presented and explained more in detail. In present form it is not clear enough. Improve all subsections 2.1, 2.2, 2.3. Add some more references.

Response 2: Thank you for pointing this out. We have incorporated the model from literature [12-15] into our work and provided proper citation and referencing. Add some more references highlighted in subsections 2.1, 2.2, 2.3. This change can be found – page 3,line107-113.

Comments 3: L100: “…computed using a Lagrange…” please explain how,  more in detail?

Response 3: Thank you for pointing this out. We have modified “... is computed using a Lagrangian that investigates the motions of each fluid particles, tracking their entire trajectory and capturing the evolution of their physical properties throughout the process [12-15]’’ . This change can be found – page 3, Line 111-113.

Comments 4:

L137-L141: insert space between number and unit, e.g. 300 mm

L161-L162: insert space between number and unit

Table 1: insert space between number and unit

L225: insert space between number and unit 80 µm

Response 4: Thank you for pointing this out. We have revised the above errors. This change can be found in the right line.

Comments 5:

Check Figure 7 value K: is capital letter correct?

Response 5: Thank you for pointing this out. Figure 7 value K should be lowercase. We have changed the errors. This change can be found in Figure 7.

Reviewer 2 Report

This paper deals with simulation of electrostatic spray painting process using an air atomizer.

Although many interesting and useful results were shown, it is insufficiency at the present stage.

Some suggestions are as follows.

Comment 1: Page 6, Line 222~

You show four figures (Figure 5 (a) – (d)) of different conditions in Figure 5. However, you explain the results only for Figure 5 (a) and (c). The results of Figure 5 (b) and (d) should be explained in the text.

Comment 2: Page 7, Line 244~

You show four figures (Figure 6 (a) – (d)) of different conditions in Figure 6. However, you do not explain each individual result in the text. The results should be explained in the text.

Comment 3: Page 9, Line 278~

You show four figures (Figure 8 (a) – (d)) of different conditions in Figure 8. However, you do not explain each individual result in the text. The results should be compared in the text.

Comment 4: Page 9, Line 292~

You show many data in Figure 9. However, you do not fully explain the results in the text. The differences between the results along X axis and Y axis should be explained in the text.

You show four experiment results and four simulation results in Figure 9. The comparison on the results among the four conditions should be discussed in the text.

Author Response

Response to Reviewer 2 Comments

1. Summary

Thank you very much for taking the time to review this manuscript. Please find the detailed responses below and the corresponding revisions in the re-submitted files.

2. Questions for General Evaluation

Reviewer’s Evaluation

Response and Revisions

Does the introduction provide sufficient background and include all relevant references?

Yes/Can be improved/Must be improved/Not applicable

We have given corresponding response in the point-by-point response letter. The same as below.

Are all the cited references relevant to the research?

Yes/Can be improved/Must be improved/Not applicable

Is the research design appropriate?

Yes/Can be improved/Must be improved/Not applicable

Are the methods adequately described?

Yes/Can be improved/Must be improved/Not applicable

Are the results clearly presented?

Yes/Can be improved/Must be improved/Not applicable

Are the conclusions supported by the results?

Yes/Can be improved/Must be improved/Not applicable

3. Point-by-point response to Comments and Suggestions for Authors

 Comment 1: Page 6, Line 222~

You show four figures (Figure 5 (a) – (d)) of different conditions in Figure 5. However, you explain the results only for Figure 5 (a) and (c). The results of Figure 5 (b) and (d) should be explained in the text.

Response 1: Thank you for pointing this out. We agree with this comment. Therefore, We have revised the manuscript. This change can be found page 7, lines248-256.

Comment 2: Page 7, Line 244~

You show four figures (Figure 6 (a) – (d)) of different conditions in Figure 6. However, you do not explain each individual result in the text. The results should be explained in the text.

Response 2: Thank you for pointing this out. We agree with this comment. Therefore, We have revised the manuscript. This change can be found page 8, lines263-266

 Comment 3: Page 9, Line 278~

You show four figures (Figure 8 (a) – (d)) of different conditions in Figure 8. However, you do not explain each individual result in the text. The results should be compared in the text.

Response 3: Thank you for pointing this out. We agree with this comment. Therefore, We have revised the manuscript. This change can be found page 9, lines302-304

Reviewer 3 Report

The manuscript entitled “Paint Film Formation Characteristics on Conical Surfaces for Electrostatic Air Spray painting” mainly focuses on the investigation and representation of the paint film formation characteristics on conical surfaces.

I think that the obtained results of this research are important and valuable.

This manuscript is well written, with a formulated key problem of the corresponding research object. Nevertheless, I have a few comments on several aspects.

An abbreviation of CFD theory was found in the text. An explanation is required.

There are some grammatical errors with this sentence: "The basic geometry of the electrostatic sprayer cap used in this study is similar to is a pneumatic atomizer, except that there is a length of 30mm electrode needle with a diameter of 0.5mm in the center of the sprayer, shown in Figure 1."

In this work, I missed a detailed characterization of the surface covered with paint, because its smoothness or roughness also significantly affects the quality of the corresponding painting.

Author Response

Response to Reviewer 3 Comments

1. Summary

Thank you very much for taking the time to review this manuscript. Please find the detailed responses below and the corresponding revisions in the re-submitted files.

2. Questions for General Evaluation

Reviewer’s Evaluation

Response and Revisions

Does the introduction provide sufficient background and include all relevant references?

Yes/Can be improved/Must be improved/Not applicable

We have given corresponding response in the point-by-point response letter. The same as below

Are all the cited references relevant to the research?

Yes/Can be improved/Must be improved/Not applicable

Is the research design appropriate?

Yes/Can be improved/Must be improved/Not applicable

Are the methods adequately described?

Yes/Can be improved/Must be improved/Not applicable

Are the results clearly presented?

Yes/Can be improved/Must be improved/Not applicable

Are the conclusions supported by the results?

Yes/Can be improved/Must be improved/Not applicable

3. Point-by-point response to Comments and Suggestions for Authors

Comments 1: An abbreviation of CFD theory was found in the text. An explanation is required.

Response 1: Thank you for pointing this out. We agree with this comment. Therefore, we have changed manuscript “computational fluid dynamics (CFD)”. This change can be found – page 2,Line51.

Comments 2: There are some grammatical errors with this sentence: "The basic geometry of the electrostatic sprayer cap used in this study is similar to is a pneumatic atomizer, except that there is a length of 30mm electrode needle with a diameter of 0.5mm in the center of the sprayer, shown in Figure 1."

Response 2: Thank you for pointing this out. We agree with this comment. Therefore, we have changed manuscript. This change can be found – page 4,Line155.

Comments 3: In this work, I missed a detailed characterization of the surface covered with paint, because its smoothness or roughness also significantly affects the quality of the corresponding painting.

Response 3: Agree. The effect of smoothness or roughness of the surface covered on the quality corresponding painting will be discussed in other research.

Reviewer 4 Report

·       Lines 78-84 – Provide citations/references for Gauss curvature. Elaborate where able.

·       Lines 85-90 – Describe the contents of the paper in a casual manner, not a list of sections.

·       110 & 111 – Provide citations for equations and descriptions.

·       128 & 129 – Provide citations for equations and descriptions.

·       Figures 1 & 2 – Citations need to be provided unless the images were created by the authors.

·       Line 173 – Provide the author of Reverence #8, don’t state according to reference 8.

·       Line 292 – Provide details of the Film Thickness Gauge – Brand, precision, applicable range of thickness.

·       Line 310 – “In the paper, the numerical…” Rephrase. Don’t refer to the paper itself.

Needs more citations, 19 for a 10-page paper isn’t a lot of support, given how many equations, approaches, models, and effects mentioned. 

Author’s need to elaborate and explain processes and methods with more depth, including which film thickness gauge was used, and the details of the laws and theories applied to the models.

Several instances of incorrect English and spelling which needs addressed. 

Overall English needs addressed. 

Author Response

Response to Reviewer 4 Comments

1. Summary

Thank you very much for taking the time to review this manuscript. Please find the detailed responses below and the corresponding revisions.

2. Questions for General Evaluation

Reviewer’s Evaluation

Response and Revisions

Does the introduction provide sufficient background and include all relevant references?

Yes/Can be improved/Must be improved/Not applicable

We have given corresponding response in the point-by-point response letter. The same as below.

Are all the cited references relevant to the research?

Yes/Can be improved/Must be improved/Not applicable

Is the research design appropriate?

Yes/Can be improved/Must be improved/Not applicable

Are the methods adequately described?

Yes/Can be improved/Must be improved/Not applicable

Are the results clearly presented?

Yes/Can be improved/Must be improved/Not applicable

Are the conclusions supported by the results?

Yes/Can be improved/Must be improved/Not applicable

3. Point-by-point response to Comments and Suggestions for Authors

Comments 1:

Lines 78-84 – Provide citations/references for Gauss curvature. Elaborate where able.

Lines 85-90 – Describe the contents of the paper in a casual manner, not a list of sections.

110 & 111 – Provide citations for equations and descriptions.

128 & 129 – Provide citations for equations and descriptions.

Response 1: Thank you for pointing this out. we have added references for Gauss curvature and elaborate it. This change can be found – page 2, Lines 82-85. We have deleted the contents of the paper on Lines 85-90. We have provided citations for equations and descriptions on lines 123 & 124, line 141 & 142.

Comments 2:

Figures 1 & 2 – Citations need to be provided unless the images were created by the authors.

Line 173 – Provide the author of Reverence #8, don’t state according to reference 8.

Line 292 – Provide details of the Film Thickness Gauge – Brand, precision, applicable range of thickness.

Response 2:

Thank you for pointing this out. We have added a physical picture of the sprayer in Figure 1 and explained the model in Figure 1 and Figure 2.We have provided the author of Reverence #8 on line. We have provided the film thickness gauge picture including the brand, precision in page 4, Lines 159.

Comments 3:

Line 310 – “In the paper, the numerical…” Rephrase. Don’t refer to the paper itself.

 Needs more citations, 19 for a 10-page paper isn’t a lot of support, given how many equations, approaches, models, and effects mentioned.

Author’s need to elaborate and explain processes and methods with more depth, including which film thickness gauge was used, and the details of the laws and theories applied to the models.

Response 3

Thank you for pointing this out. The contents on Line 310 have been revised. And we have added more citations in this manuscript. We also have added explained on the processes and methods of the research. This change can be found in page3,lines107-108, 111-113, and page 10, lines310-312.

Comments on the Quality of English Language

4. Response to Comments on the Quality of English Language

Point 1: Several instances of incorrect English and spelling which needs addressed.

Overall English needs addressed.

Response4: We have revised some incorrect English and spelling, and overall English grammars have been changed.

Reviewer 5 Report

The subject of this manuscript, “Paint Film Formation Characteristics on Conical Surfaces for Electrostatic Air Spray painting”, is within the scope of Coatings.  However, in its current state the manuscript contains limitations. Additional work is needed to convert this into a full-fledged manuscript.

General comments:

  • The main problem with this manuscript is that parts of it read more like a technical report and lack the necessary quality for a research article. For example, none of the 19 provided references are cited in the “4 Results and Discussion” and “5 Experimental Investigation” sections; a more detailed discussion of the results (including a comparison to available results from literature) is required.
  • Be consistent, use gaps between numbers and units (e.g. “250 mm”, rather than “250mm”), use “MPa” rather than “Mpa”…

INTRODUCTION: The authors cite 16 references in this section, leading the reader the purpose of their work.

RESULTS AND DISCUSSION & EXPERIMENTAL INVESTIGATION:  As mentioned above, a more detailed discussion is required in these sections (comparison to existing literature…).

FIGURES:  Please

·       enlarge Fig. 7 (or at least increase the font size of the text on the axes) and correct the text on the axes (change “paint transfer efficiency(%)” to “Paint transfer efficiency (%)” and “CurvatureK” to “Curvature k”?).

·       include scale bars in Fig. 8(a)-(d).

·       increase font sizes in Fig. 9.

REFERENCES: Ref. [17] is not cited in the manuscript text.

To summarise, this could be an interesting study. However, more work is needed at this stage before it can be considered for publication.

Moderate editing of English language required.

Author Response

Response to Reviewer 5 Comments

1. Summary

Thank you very much for taking the time to review this manuscript. Please find the detailed responses below and the corresponding corrections highlighted.

2. Questions for General Evaluation

Reviewer’s Evaluation

Response and Revisions

Does the introduction provide sufficient background and include all relevant references?

Yes/Can be improved/Must be improved/Not applicable

We have given corresponding response in the point-by-point response letter. The same as below

Are all the cited references relevant to the research?

Yes/Can be improved/Must be improved/Not applicable

Is the research design appropriate?

Yes/Can be improved/Must be improved/Not applicable

Are the methods adequately described?

Yes/Can be improved/Must be improved/Not applicable

Are the results clearly presented?

Yes/Can be improved/Must be improved/Not applicable

Are the conclusions supported by the results?

Yes/Can be improved/Must be improved/Not applicable

3. Point-by-point response to Comments and Suggestions for Authors

Comments 1:

General comments:

       The main problem with this manuscript is that parts of it read more like a technical report and lack the necessary quality for a research article. For example, none of the 19 provided references are cited in the “4 Results and Discussion” and “5 Experimental Investigation” sections; a more detailed discussion of the results (including a comparison to available results from literature) is required.

Response1: Thank you for pointing this out. The research on the effect of surface characteristics, including shape and curvature, on paint deposition and paint transfer efficiency has not been modeled and investigated extensively, especially for conical surface. And we still have added a more detailed discussion of the results in Results and Discussion. This change can be found in page 11, lines 325-333.

Comments 2:

       Be consistent, use gaps between numbers and units (e.g. “250 mm”, rather than “250mm”), use “MPa” rather than “Mpa”…

Response 2: Agree. We have, accordingly, changed the manuscript in in corresponding section.

Comments 3:

   INTRODUCTION: The authors cite 16 references in this section, leading the reader the purpose of their work.

RESULTS AND DISCUSSION & EXPERIMENTAL INVESTIGATION:  As mentioned above, a more detailed discussion is required in these sections (comparison to existing literature…).

Response 3: Thank you for pointing this out. we have added a more detailed discussion of the results in Results and Discussion. This change can be found in page9, lines 302-303,page 11, lines 325-333.

Comments 4:

FIGURES:  Please enlarge Fig. 7 (or at least increase the font size of the text on the axes) and correct the text on the axes (change “paint transfer efficiency(%)” to “Paint transfer efficiency (%)” and “CurvatureK” to “Curvature k”?).

     include scale bars in Fig. 8(a)-(d).

increase font sizes in Fig. 9.

REFERENCES: Ref. [17] is not cited in the manuscript text.

Response4: Thank you for pointing this out. We have changed Fig. 7, Fig. 8(a)-(d) and Fig. 9, Ref. [17] has been cited in the manuscript text.

Round 2

Reviewer 2 Report

This article was revised enough, so it can be published.

Author Response

Thank you very much for taking the time to review this manuscript. Thanks for all your comments. 

Reviewer 5 Report

Although the authors have improved their manuscript, more work is still needed before it can be accepted in Coatings. 

The main problem with this manuscript is the lack of a detailed discussion of the results in the ”4 Results and Discussion” and “5 Experimental Investigation” sections. Only one reference [9] is cited in the “5 Experimental Investigation” section, NO references are cited in the”4 Results and Discussion” section. A comparison of the results to available data from literature is required.

INTRODUCTION: The authors cite now 18 references in this section, leading the reader the purpose of their work.

RESULTS AND DISCUSSION & EXPERIMENTAL INVESTIGATION:  As mentioned above, a more detailed discussion is still required in these sections (comparison to available results from literature…).

The FIGURES have been improved.

Change “250mm” to “250 mm” in line 281.

Overall, 20 REFERENCES are now cited in the manuscript text.

To summarise, there is still more work needed at this stage before it can be considered for publication.

Author Response

Response to Reviewer 5 Comments

1. Summary

Thank you very much for taking the time to review this manuscript. Please find the detailed responses below and the corresponding corrections highlighted.

2. Questions for General Evaluation

Reviewer’s Evaluation

Response and Revisions

Does the introduction provide sufficient background and include all relevant references?

Yes/Can be improved/Must be improved/Not applicable

We have given corresponding response in the point-by-point response letter. The same as below

Are all the cited references relevant to the research?

Yes/Can be improved/Must be improved/Not applicable

Is the research design appropriate?

Yes/Can be improved/Must be improved/Not applicable

Are the methods adequately described?

Yes/Can be improved/Must be improved/Not applicable

Are the results clearly presented?

Yes/Can be improved/Must be improved/Not applicable

Are the conclusions supported by the results?

Yes/Can be improved/Must be improved/Not applicable

3. Point-by-point response to Comments and Suggestions for Authors

Comments 1:

General comments:

       The main problem with this manuscript is the lack of a detailed discussion of the results in the ”4 Results and Discussion” and “5 Experimental Investigation” sections. Only one reference [9] is cited in the “5 Experimental Investigation” section, NO references are cited in the”4 Results and Discussion” section. A comparison of the results to available data from literature is required.

RESULTS AND DISCUSSION & EXPERIMENTAL INVESTIGATION:  As mentioned above, a more detailed discussion is still required in these sections (comparison to available results from literature…).

Response1: Thank you for pointing this out. we have cited some references for comparison of the results and discussion of the results in Results and Discussion section. This change can be found in page 6, lines 203-206; page 6, lines 218-221; page 7, lines 235-240 and page 7, lines 259-262,

Comments 2:

       Change “250mm” to “250 mm” in line 281.

Response 2: Thank you for pointing this out. We have added space in the manuscript text.